# In Vitro and Preclinical Antitumor Evaluation of Doxorubicin Liposomes Coated with a Cholesterol-Based Trimeric β-D-Glucopyranosyltriazole

**DOI:** 10.3390/pharmaceutics15122751

**Published:** 2023-12-11

**Authors:** Aline Teixeira Maciel e Silva, Ana Luiza Chaves Maia, Juliana de Oliveira Silva, Sued Eustáquio Mendes Miranda, Talia Silva Cantini, Andre Luis Branco de Barros, Daniel Crístian Ferreira Soares, Mariana Torquato Quezado de Magalhães, Ricardo José Alves, Gilson Andrade Ramaldes

**Affiliations:** 1Departamento de Produtos Farmacêuticos, Faculdade de Farmácia, Universidade Federal de Minas Gerais, Av. Presidente Antônio Carlos, 6627, Belo Horizonte 31270-901, MG, Brazil; aline_tms@yahoo.com.br (A.T.M.e.S.); ana.chavesmaia@gmail.com (A.L.C.M.); julianaoliveira.far@gmail.com (J.d.O.S.); sued1989@gmail.com (S.E.M.M.); taliacantini@gmail.com (T.S.C.);; 2Departamento de Análises Clínicas e Toxicológicas, Faculdade de Farmácia, Universidade Federal de Minas Gerais, Av. Presidente Antônio Carlos, 6627, Belo Horizonte 31270-901, MG, Brazil; 3Laboratório de Bioengenharia, Universidade Federal de Itajubá, Rua Irmã Ivone Drumond, 200, Distrito Industrial II, Itabira 35903-087, MG, Brazil; soares@unifei.edu.br; 4Departamento de Bioquímica e Imunologia, Instituto de Ciências Biológicas, Universidade Federal de Minas Gerais, Av. Presidente Antônio Carlos, 6627, Belo Horizonte 31270-901, MG, Brazil; mquezado@icb.ufmg.br

**Keywords:** liposomes, surface coating, carbohydrates, antitumor activity

## Abstract

The coating of liposomes with polyethyleneglycol (PEG) has been extensively discussed over the years as a strategy for enhancing the in vivo and in vitro stability of nanostructures, including doxorubicin-loaded liposomes. However, studies have shown some important disadvantages of the PEG molecule as a long-circulation agent, including the immunogenic role of PEG, which limits its clinical use in repeated doses. In this context, hydrophilic molecules as carbohydrates have been proposed as an alternative to coating liposomes. Thus, this work studied the cytotoxicity and preclinical antitumor activity of liposomes coated with a glycosyl triazole glucose (GlcL-DOX) derivative as a potential strategy against breast cancer. The glucose-coating of liposomes enhanced the storage stability compared to PEG-coated liposomes, with the suitable retention of DOX encapsulation. The antitumor activity, using a 4T1 breast cancer mouse model, shows that GlcL-DOX controlled the tumor growth in 58.5% versus 35.3% for PEG-coated liposomes (PegL-DOX). Additionally, in the preliminary analysis of the GlcL-DOX systemic toxicity, the glucose-coating liposomes reduced the body weight loss and hepatotoxicity compared to other DOX-treated groups. Therefore, GlcL-DOX could be a promising alternative for treating breast tumors. Further studies are required to elucidate the complete GlcL-DOX safety profile.

## 1. Introduction

Cancer is one of the most common causes of death worldwide, and breast tumors are the most incident type in women [1]. Despite the development of different therapeutic approaches, Doxorubicin (DOX), an anthracycline used as a chemotherapeutic agent, is still the choice drug for treating diverse types of cancers including lung, thyroid, ovarian, and, in particular, breast tumor malignancies [2,3,4]. This drug’s major inconvenience is the severe toxic effects observed in many patients, where cardiotoxicity occupies a relevant position. Seeking to overcome these problems, many nanosystems loaded with DOX have been proposed in recent decades, aiming to improve the pharmacokinetics profile, while the reduction in drug toxicity is also desired [5,6,7,8].

An example of the revolutionary role of science in this field can be attributed to the development of Doxil^®^, which was approved for clinical use in 1995. Doxil^®^ is a liposomal formulation with an intrinsic surface modification conducted by adding polyethyleneglycol (PEG) molecules. This modification confers to the liposomal vesicles a superior drug residence time in the blood circulation, in comparison to conventional liposomes, and significantly increases their tumor accumulation via passive targeting mediated through the well-known EPR effect (enhanced permeability and retention) [9,10,11].

More recently, studies have shown the disadvantages of the PEG molecule as a long-circulation agent. Among them, the most important is the immunogenic role of PEG, observable after being administrated in different time intervals. Notably, PEG-liposomes display a significant reduction in the circulation time and increase uptake in hepatic and splenic tissues after accumulated administrations [12,13]. This phenomenon is known as the Accelerated Blood Clearance (ABC) effect, and it is characterized by the production of IgM antibodies (anti-PEG) as a response to the first PEG-liposome administration. The produced antibodies are capable of binding to the PEG polymer administrated in the second dose, activating the complement system and resulting in an important increment of PEG-liposome uptake in the Kupffer cells [14,15,16].

Still, in 2012, Suzuki and collaborators demonstrated that the ABC effect occurred in beagle dogs after the recurrent administration of Doxil^®^ [12]. In the same year, similar data were found by Li and co-workers, who reported an ABC effect in mice, rats, and also beagle dogs [17]. Other studies revealed that the ABC effect varies in different mammalian species, and a more pronounced effect is observed when the dose used is less than 2 mg/m^2^ of DOX [17,18]. In humans, clinical evaluations conducted by Chanan-Khan and collaborators (2003) revealed that Doxil^®^ could activate the complement system of twenty-one patients from a total of twenty-nine individuals studied. Furthermore, the authors observed hypersensibility reactions in thirteen patients from twenty-nine individuals studied [19].

Seeking to find alternatives to the PEG, many studies have been proposing the use of biodegradable polymers, polyvinyl alcohols, amphiphilic vinyl-pyrrolidones, and soluble polymers such as HPMA, PVP, PMOX, and PDMA [20,21,22,23]. On the other hand, promising alternatives have been evaluated using hydrophilic natural molecules such as carbohydrates. A pioneering study considering the stealth properties of carbohydrates coating liposomes was conducted by Pain and collaborators in 1984 [24]. However, only ten years later, in 1994, the study conducted by Maruyama and co-workers prepared liposomes loaded with DOX and coated with the monosialoganglioside GM1. The study revealed that the proposed surface coating could induce a similar behavior compared to PEGylated liposomes. In this study, the authors observed that after 6 h, the DOX concentration in the bloodstream was 2.3 and 2.9 times higher for the liposomes constituted by GM1 and PEG, respectively, compared to conventional liposomes [25].

Considering the necessity to develop new formulations with a high immune-compatible profile, this work studied the in vitro and preclinical antitumor activity of liposomes coated with a glycosyltriazole glucose derivative as a potential strategy against breast cancer.

## 2. Materials and Methods

DOX was supplied by Lancrix Chemicals (Shanghai, China), with a purity higher than 99.9%. Hydrogenated soy phosphatidylcholine (HSPC), egg phosphatidylglycerol (EPG), and Distearoyl-phosphoethanolamine-polyethilenglycol2000 (DSPE-PEG2000) were acquired from Lipoid GmbH (Ludwigshafen, Germany). Cholesterol (CHOL) and HEPES salt were supplied by Sigma Chemical Company (St. Louis, MO, USA). The polycarbonate membrane was purchased from Millipore (Billerica, MA, USA). All other compounds and solvents used in this work were of analytical grade. RPMI 1640 Medium, fetal bovine serum (FBS), penicillin, streptomycin, and trypsin EDTA 0.25% were purchased from Gibco-Invitrogen (Waltham, MA, USA). The subcutaneous tumor model was established in 7–8-week female BALB/c mice purchased from CEBIO-UFMG (Belo Horizonte, Brazil).

### 2.1. Synthesis of the Glycosyltriazole Glucose

In a previous study conducted by Maciel e Silva et al. (2018), a series of new glycosyltriazole compounds derived from D-galactose and *N*-acetylglucosamine were synthesized, seeking to prepare carbohydrates with stealth properties for biomedical applications [26]. The authors, in particular, synthesized a trimeric glycosyltrizole (TGT-30) derived from the D-glucose. Aiming to produce liposomes, the obtained compound was synthetically attached to the 3-*O*-(2-aminoetil)cholesterol molecule, followed by a deacetylation reaction [27]. The set of reactions allows for the obtention of the cholesterol conjugate, designated as CHO-TGT-30, in which the molecular structure is available in Figure 1. The detailed scheme of the synthesis and characterization of CHO-TGT-30 is presented in the Appendix A.

### 2.2. Liposome Preparation

Chloroform aliquots of HSPC:EPG:CHO-TGT-30: or HSPC:EPG:CHOL:DSPE:PEG, in a molar ratio of 11.20:2.16:6.89:1.137, were added to a round-bottom flask, and the chloroform was removed under a reduced pressure. The formulations prepared were designated as GlcL and PegL for the vesicles containing the CHO-TGT-30 and DSPE:PEG compounds, respectively. Then, the two liposome films were hydrated with a 300 mM ammonium sulfate solution (pH 7.4) at 25 °C [28]. The obtained liposomes were calibrated by extrusion with polycarbonate membranes of 0.4 μm, 0.2 μm, and 0.1 μm, 10 cycles per membrane, using the Lipex Biomembranes extruder, model T001 (Vancouver, BC, Canada). Afterward, non-encapsulated ammonium sulfate was removed by ultracentrifugation (Ultracentrifuge Optima^®^ L-80XP, Beckman Coulter, Brea, CA, USA) at 150,000× *g* and 4 °C for two hours, and the pellet was resuspended with HEPES saline buffer pH 7.4. Then, 2 mg of DOX was dissolved in 1 mL of GlcL or PegL dispersion, and the mixture was kept for 2 h at 60 °C for drug encapsulation. The DOX in the external medium was purified by ultracentrifugation, as previously described.

### 2.3. Physicochemical Characterization

The Z-average diameter and polydispersity index (PDI) of the produced liposomes were determined through the dynamic light scattering (DLS), evaluated at 25 °C and an angle of 90°. The zeta potential was determined by electrophoretic mobility in conjunction with DLS. Analyses were carried out using the NanoZS90 zetasizer (Malvern Instruments, Malvern, England). All samples were analyzed in triplicate after dilution with HEPES salt buffer pH 7.4 at a ratio of 1:30.

### 2.4. Encapsulation Efficiency Evaluation

DOX was quantified through the HPLC technique. The method employed used a 4.6 mm × 250 mm, 5 μm ACE^®^ C_8_ column (ACE, Aberdeen, UK) eluted at 25 °C, with a mobile phase constituted by methanol:phosphate buffer at pH 3.0 (65:35) using a flow rate of 1.0 mL/min (injection volume of 20 μL). The elution products were excited and detected at 470 and 555 nm, respectively. A calibration curve was built using five points ranging from 1.0 to 0.0625 μg/m. For the quantification of DOX in liposomal formulations, vesicles were opened to access the drug content by using isopropyl alcohol in a ratio of 1:2. The concentration of doxorubicin was measured before and after ultracentrifugation, and the Encapsulation Percentage (EP) was calculated using the equation below:EP (%) = ([DOX] centrifuged liposomes/[DOX] non-centrifuged liposomes) × 100

### 2.5. Storage Stability and Drug Release Profile Evaluation

Immediately after each preparation, the liposomes were stored at 4 °C in an Ambar flask and under N_2_ atmosphere. At 7; 15; 30; 60; 90; 120; and 180 days post-preparation, the liposomes were characterized in terms of the Z-average diameter, PDI, zeta potential, and encapsulation percentage (EP).

The DOX release profile was performed by the dialysis method using cellulose membranes with a cutoff size of 14 kDa and a diameter of 21 mm (cellulose ester membrane; Sigma–Aldrich, São Paulo, Brazil). An aliquot of 1 mL of the formulation was transferred to the dialysis bags. The bags were sealed and incubated with 100 mL of PBS (pH 7.4) at 37 °C, for 48 h, under stirring at 150 rpm. The aliquots were withdrawn, the same volume was replaced, the DOX concentration was analyzed by HPLC, and the values were plotted as the cumulative percentage of drug release.

### 2.6. Cell Culture and Cytotoxicity Study

In this study, the 4T1 metastatic mice breast tumor cell line (CRL-2539–ATCC) was grown in RPMI 1640 medium, supplemented with 10% (*v*/*v*) of FBS, penicillin (100 IU/mL), and streptomycin (100 μg/mL). The cells were kept in a humidified atmosphere containing 5% CO_2_ at 37 °C. For the cytotoxicity assay, 4T1 cells were seeded in 96-well plates (1 × 10^4^ cells/well) at 24 h before treatment. The cells were then exposed to different concentrations of free-DOX, GlcL-DOX, PegL-DOX, and blank liposomes for 48 h.

The cytotoxic activity was assessed, in triplicate, by the 3-(4,5-dimethyl-2-thyazolyl)-2,5-diphenyltetrazolium bromide colorimetric assay (MTT, Sigma-Aldrich, São Paulo, Brazil). Colorimetric measurements were performed at 550 nm, using the SpectraMax M5e microplate reader (Molecular Devices, San Jose, CA, USA). The results were evaluated using the software OriginPro 8.0 and expressed by IC_50_ values.

Seeking to evaluate the cytotoxicity mechanisms, the 4T1 cells were evaluated through the Flow Cytometry Technique, using the LSR Fortessa cytometer (BD Biosciences, San Jose, CA, USA). Thus, the cells were treated with free-DOX, GlcL-DOX, or PegL-DOX at concentrations corresponding to the IC_50_ value, previously determined through an MTT assay. After the treatment of 48 h, the cells were centrifuged at 1100 rpm and resuspended in a culture medium. For the quantification of cells undergoing early and late apoptosis, as well as necrosis, the Annexin V-Alexa Fluor 488 Apoptosis Detection Kit (BD Biosciences, San Jose, CA, USA) was used. The cells were stained with 2.5 μL of Alexa fluor 488-conjugated Annexin-V and 5 μL of propidium iodide (PI) solution for 10 min, at room temperature, and protected from light. Then, the cells were evaluated by a flow cytometer (50,000 events per sample).

### 2.7. Preclinical Evaluation of Antitumor Activity

Female BALB/c mice (~22 g) were supplied by CEBIO-UFMG (Belo Horizonte, Brazil). Aliquots of 1.0 × 10^6^ of 4T1 cells (100 μL) were administrated subcutaneously into the right flanks of the mice. At 7 days post-implantation, a palpable tumor of around 100 mm^3^ was achieved. Animals have free access to standard food and water. All the animal assays were performed following the National Research Council’s Guide for the Care and Use of Laboratory Animals (NIH-USA) and were approved by the local Ethics Committee for Animal Experiments (CEUA/UFMG) under protocol #158/2014.

Mice were randomly divided into four different groups (*n* = 7 per group), nominated as: (1) Control (Saline 0.9% *w*/*v*); (2) Free-DOX; (3) GlcL-DOX; and (4) PegL-DOX. The injection was performed intravenously by the tail vein with a dose of DOX of 5 mg/kg/day, in a total of four administrations. The accumulative dose of DOX reached was 20 mg/kg. During the whole experiment, the tumors were measured with a caliper every 2 days, and the mice’s body weights were also determined. The tumor volumes were calculated from the following formula, where d_1_ and d_2_ represent the smaller and larger diameter, respectively [29]:V = (d_1_)^2^ × d_2_ × 0.5

At four days after the last treatment dose (Day 19), the mice were euthanized under anesthesia (ketamine-80 mg/kg and xylazine-15 mg/kg). The relative tumor volume (RTV) and the tumor growth inhibition ratio (IR) were determined by the following formulas [29]:$$\mathrm{RTV}=\frac{\mathrm{Tumor}\;\mathrm{volume}\;\mathrm{on}\;\mathrm{day}\;19\;\mathrm{after}\;\mathrm{implant}}{\mathrm{Tumor}\;\mathrm{volume}\;\mathrm{on}\;\mathrm{day}\;7\;\mathrm{after}\;\mathrm{implant}}$$
$$\mathrm{IR}=1-\frac{\mathrm{Mean}\;\mathrm{RTV}\;\mathrm{from}\;\mathrm{each}\;\mathrm{treatment}}{\mathrm{Mean}\;\mathrm{RTV}\;\mathrm{from}\;\mathrm{control}\;\mathrm{group}}\times 100$$

Blood samples were collected at the end of the treatment and centrifuged at 3500 rpm for 10 min for plasma collection. Biochemical analyses were carried out to assess the potential toxic effects of treatments. The plasmatic urea, creatinine, AST (aspartate aminotransferase), ALT (alanine aminotransferase), and CK-MB were measured. The biochemical tests were performed using commercial kits from Labtest^®^ (Lagoa Santa, Brazil) through Bioplus BIO-2000 semiautomatic analyzer equipment (São Paulo, Brazil).

### 2.8. Statistical Analysis

The results were presented as the mean ± SD. GraphPad PRISM, version 6.00 software (GraphPad Software Inc., La Jolla, CA, USA), was used for statistical analysis. The data were evaluated by one-way analysis of variance (ANOVA), followed by the Tukey test, or *t*-test, when the number of groups evaluated was equal to two. All data showed a normal distribution and homoscedasticity, when necessary. *p*-values lower than 0.05 indicate statistically significant differences between the groups.

## 3. Results

### 3.1. Liposome Characterization

The results of the liposome characterization are described in Table 1. Both liposomes showed a suitable size for intravenous administration, smaller than 200 nm, with a low PDI (<0.3), indicating a monodisperse formulation.

The zeta potential values for both liposomes were close to neutrality, which might be explained by the impaired electrophoretic mobility due to the coverage of the vesicles with PEG or carbohydrates. Although a zeta potential over ±30 mV is aimed for to avoid the fusion of the vesicles by electrostatic repulsion, the presence of the PEG chains or the carbohydrate onto the surface of the liposomes leads to steric repulsion between the vesicles. Thus, aggregation, size modification, and the release of the encapsulated liposome content are inhibited [30]. Additionally, both liposomes showed a great ability to encapsulate the drug, as expected, by using the ammonium sulfate gradient approach [31].

### 3.2. Storage Stability Evaluation

The liposomes were stored for 180 days at 4 °C, and the Z-Average diameter, PDI, zeta potential, and EP over time are shown in Figure 2. It is noteworthy that the liposomes showed long-term stability throughout the study, as indicated by a stable mean diameter, PDI, and zeta potential. Slight variations in the encapsulation percentage were observed for both liposomes, where the pegylated liposomes showed a decrease in EP of 23 ± 11%, while in the carbohydrate-coated liposome, a comparable decrease in EP (16 ± 9%) was observed over time. However, only PEGylated liposomes showed a statistically significant decrease in EP, comparing days 0, 60, and 90 after preparation (indicated by an asterisk in Figure 2D). This finding suggests the better stability of GlcL-DOX over PegL-DOX.

### 3.3. Drug Release Profile

Figure 3 shows the release profile of the GlcL-DOX and PegL-DOX. The data revealed the ability of both vesicles to release doxorubicin slowly and maintain stability after an initial burst during the 48 h evaluated. On the other hand, liposomes presented slightly different release behavior, in which the formulation containing the carbohydrate on their surface demonstrated a higher drug release (around 30%) compared to PEGylated liposomes (around 20%) during 48 h.

### 3.4. Cytotoxicity Assay and Cell Death Profile

Table 2 shows the IC_50_ data for free-DOX, GlcL-DOX, and PegL-DOX. It was observed that the carbohydrate-coated liposome statistically reduced the cell viability in comparison with the PEGylated liposome. Although a higher IC_50_ was observed for liposomes compared to the free drug, it is important to underscore that GlcL-DOX was three times more cytotoxic than PegL-DOX. The higher DOX cytotoxic effect in free form can be related to the ability of this drug to reach the action site after cell uptake [32]. However, this same behavior has not been observed for liposomal formulations, which are reported to be initially internalized, and the release from lysosomes could be delayed [33].

Figure 4 shows flow cytometry panels (PE-Texas-Red or PI vs. Alexafluor 488) obtained after the described in vitro study. In panel A, the negative control gating is available, revealing one single population of live cells, which constituted 99.06%. The number of viable cells in all treated groups, represented by panels B, C, and D, was significantly reduced due to the employed compounds. Thus, panel B is the available data for the cell group treated with free-DOX. It is possible to observe that the cells in the early apoptosis stage were quantified at 3.8%, while in late apoptosis, they reached 44.4%. Furthermore, the percentage of necrotic cells reached around 4%. The treatments with liposomes increased in the cell population in early apoptosis, with a percentage of 53.4% for GlcL-DOX (Figure 4C) and 42.6% for PegL-DOX (Figure 4D).

### 3.5. In Vivo Antitumor Activity and Preliminary Toxicity

The antitumor activities of free-DOX, GlcL-DOX, and PegL-DOX were evaluated, and the results are shown in Figure 5. In the tumor volume evaluation, the control group showed faster tumor growth, which increased exponentially throughout the study. In the treated groups, free-DOX showed a lower capacity to control tumor growth, while liposomes could better control tumor growth. Interestingly, for GlcL-DOX, tumor growth was significantly controlled compared to the other groups.

Table 3 describes the RTV and IR data. GlcL-DOX-treated animals showed a low RTV, with a value significantly lower than that of the control, PegL-DOX, and free drug groups. Finally, the IR results showed a 58.5% inhibition of tumor growth in the GlcL-DOX-treated animals, a 35.3% inhibition in the PegL-DOX group, and a 24.3% inhibition in the free DOX-treated animals.

The preliminary toxicity of animals receiving GlcL-DOX was evaluated by body weight follow-up and biochemical analysis of the liver, kidney, and heart. Figure 6 shows the change in the body weight of animals treated with the different DOX-containing formulations and saline 0.9% (Control). Regardless of the formulation administered, there was a significant weight loss compared to the control group. Only animals in the control group showed a weight gain from day 9 (after the first dose of the treatment) until day 19 (end of the experiment). When the weight of the animals receiving the different treatments was evaluated at the end of the experiment, we found that the group treated with GlcL-DOX and PegL-DOX had a 10% higher weight than the group treated with Free-DOX. In addition, a mortality rate of 12.5% was observed in the group treated with Free-DOX, whereas no deaths were observed in the groups treated with both liposomes. These results suggest that the use of a liposomal formulation can reduce the toxicity of DOX [34,35].

Table 4 shows that the AST enzymatic activity remained unchanged in all treated groups. On the other hand, ALT levels were statistically higher in the animals treated with Free-DOX and PegL-DOX. Interestingly, the ALT activity was reduced in animals receiving GlcL-DOX compared to the drug-treated groups. Additionally, an increase in CK-MB was observed in animals treated with Free-DOX, and these levels were significantly lower in animals treated with liposomal formulations. Similar behavior was observed in urea levels; however, histological analyses are fundamental to elucidating the nephrotoxicity.

## 4. Discussion

Liposome is the most studied nanostructured pharmaceutical formulation as a feasible option for the treatment of various diseases. The surface modification of vesicles has been used, aiming to modify the pharmacokinetics profile and consequently modulate the drug delivery, drug release, and circulation time [10,36]. Among these modifications, the hydrophilic polymer PEG is extensively described as an important surface functionalization that is associated with a longer liposome circulation time, altered biodistribution, and the metabolism of encapsulated drugs [10,30]. However, recent studies showed the decreased cellular uptake and antitumor activity of PEG-functionalized liposomes [37]. In addition, the occurrence of toxicity when used intravenously in repeated doses has been observed in the clinical use of PEGylated liposomes [38,39].

Therefore, the interest in new molecules used to modify the biological distribution and extend the circulation time with minor immune system stimuli has been growing in recent years. The glycolipids have been described as promising molecules, which are associated with a prolonged circulation time and reduced uptake by the mononuclear phagocyte system (MPS) cells. In addition, the functionalization could promote the active targeting of encapsulated drugs by the tumor cell due to its higher metabolism and uptake of carbohydrates [40,41].

In this study, we proposed the functionalization of liposomal vesicles with glucose and evaluated their influence on physicochemical parameters such as the mean diameter, zeta potential, polydispersity index, and encapsulation efficiency. We also studied the storage stability of liposomes, the in vitro release of DOX, and the preclinical antitumor activity and preliminary toxicity.

It was observed that carbohydrate-decorated liposomes, as a potential substitute for PEG, showed an improvement in storage stability, especially in long-term EP. It is well-known that the steric repulsion of the hydrophilic PEG chains prevents the aggregation/fusion process contributing to the storage stability of liposomes [30,42]. Importantly, GlcL-DOX showed a similar stability profile indicating that carbohydrates can reproduce the stability properties of PEGylated formulations. These findings were very promising and encouraged further in vitro and in vivo assays.

The 4T1 cell murine breast tumor has been considered an advantageous preclinical model for researching new therapeutic strategies due to similarities with invasive breast carcinomas in morphological aspects and clinical evolution [43]. In this study, despite the modest results of cytotoxicity, the advantage of GlcL-DOX in controlling tumor growth was quite evident (Figure 5 and Table 3). In addition, the DOX systemic toxicity was reduced when encapsulated in liposomes, especially GlcL-DOX.

DOX is a first-line anticancer agent; meanwhile, side effects severely limit its clinical use. Therefore, to evaluate the preliminary safety of treatments, the biochemical parameters of liver, heart, and kidney injury were investigated. Cardiotoxicity is the most limiting in the clinical use of doxorubicin [44]. The presence of cardiac injury can be monitored by the levels of CK-MB activity in plasma. In the present study, increased CK-MB activity was observed in animals treated with free-DOX; however, these levels were significantly lower in animals treated with liposomal formulations. These results corroborate studies that showed a reduction in DOX cardiac toxicity after encapsulation in liposomes [31].

DOX-induced liver damage is a consequence of the production of radical oxygen species (ROS) and lipid peroxidation, promoting oxidative damage to cells. Once initiated, oxidative stress can reduce the regenerative capacity of liver cells. In this case, these irreversible changes lead to apoptosis or necrosis of the hepatocytes, and an increase in liver enzymes in the plasma, mainly ALT and AST, is observed [45,46]. Free DOX- and PegL-DOX-treated animals showed higher ALT plasmatic enzyme activity compared to the control group and GlcL-DOX, suggesting the hepatoprotective effect of carbohydrate-coated liposomes.

## 5. Conclusions

In this study, a new DOX-loaded carbohydrate-coated liposome (GlcL-DOX) was prepared and characterized with a suitable size, zeta potential, and storage stability for preliminary biological analyses. The in vivo study showed that glucose-coating liposomes enhanced the antitumor activity compared to the Free-DOX and PEGylated liposomes. In addition, it was observed in the biochemical analysis a great potential of GlcL-DOX to reduce the cardiac and hepatic toxicity of DOX. Therefore, the incorporation of carbohydrates on the surface of liposomes could be a promising alternative for treating breast tumors. Further studies are required to elucidate the complete GlcL-DOX safety profile.

## Figures and Tables

**Figure 1 pharmaceutics-15-02751-f001:**
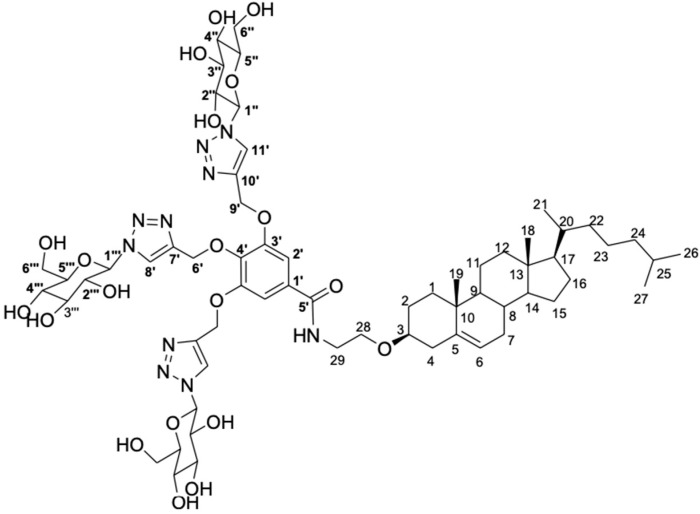
Molecular structure of trimeric glycosyltrizole carbohydrate-cholesterol (CHO-TGT-30) obtained from D-glucose and 3-O-(2-aminoetil)-cholesterol.

**Figure 2 pharmaceutics-15-02751-f002:**
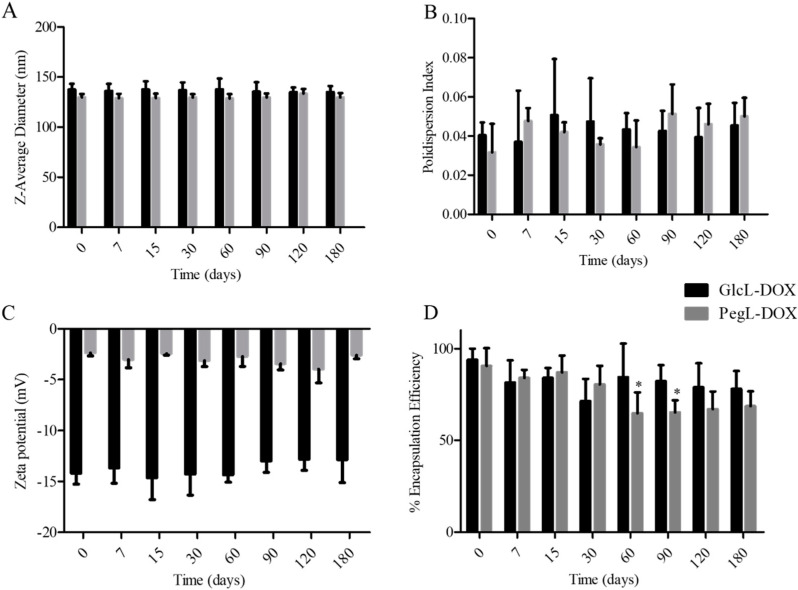
Storage stability of liposomes stored at 4 °C for 180 days. (**A**) Z-Average diameter, (**B**) Polydispersity Index (PDI), (**C**) zeta potential, and (**D**) Encapsulation Percentage (% EP). Results are presented as the mean (*n* = 3) ± SD. All data were analyzed by one-way ANOVA analysis of variance, followed by Tukey’s post-test. * represents statistical differences (*p* < 0.05) compared to day 0.

**Figure 3 pharmaceutics-15-02751-f003:**
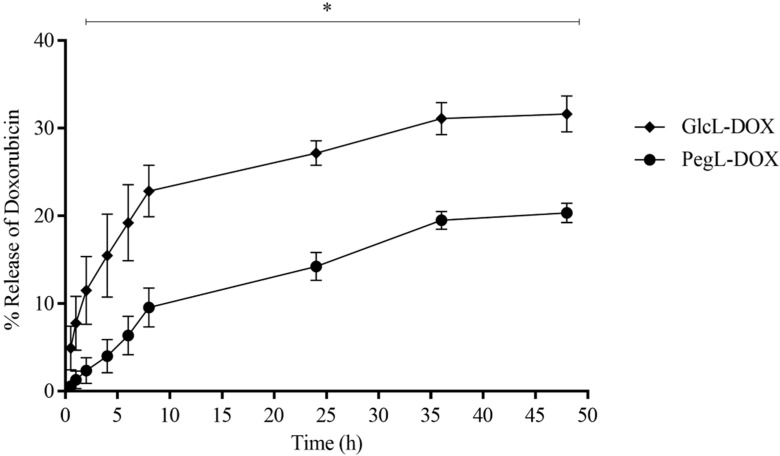
In vitro drug release profile of DOX from the liposomes GlcL-DOX and PegL-DOX at 37 °C for 48 h. Results are presented as the mean (*n* = 3) ± SD. All data were analyzed by one-way ANOVA analysis of variance, followed by Tukey’s post-test. * represents statistical differences (*p* < 0.05) between the groups.

**Figure 4 pharmaceutics-15-02751-f004:**
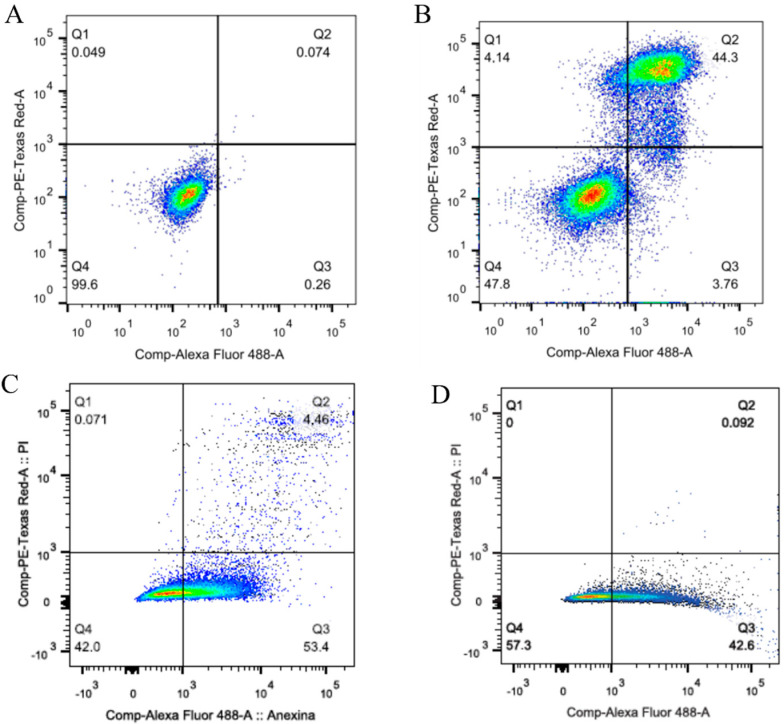
Flow cytometry panels obtained from 4T1 cells stained with Annexin-V-Alexa Fluor-488 and PI solution (PE-Texas Red-A channel) constituted by the: (**A**) Negative Control group; (**B**) cells treated with Free-DOX; and (**C**,**D**) cells treated with GlcL-DOX and PegL-DOX formulations, respectively.

**Figure 5 pharmaceutics-15-02751-f005:**
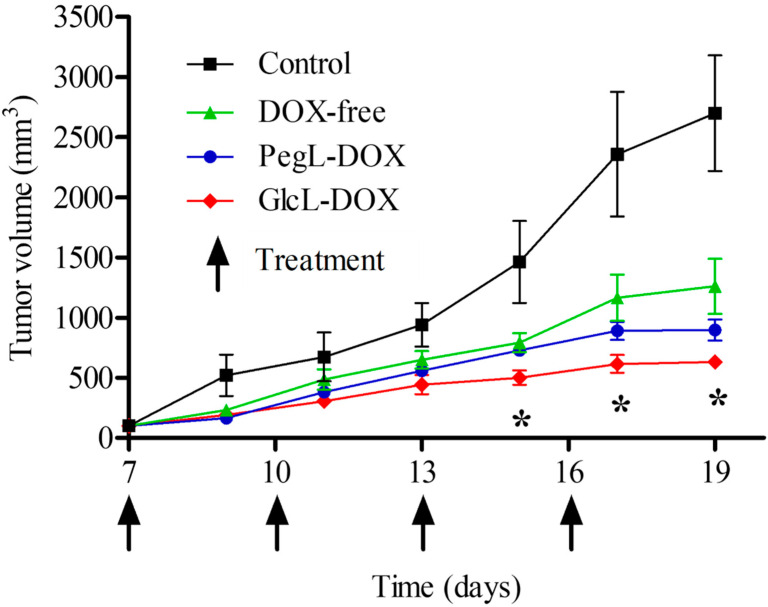
Antitumor activity of Saline (Control), Free-DOX, GlcL-DOX, and PegL-DOX against a 4T1 breast tumor implanted in the flank of female BALB/c mice. Each treatment was intravenously administered four times, every 2 days, at a dose of 5 mg/kg. Results are presented as the mean ± SD (*n* = 7). All data were analyzed by one-way ANOVA analysis of variance, followed by Tukey’s post-test. * represents statistical differences between GlcL-DOX and PegL-DOX formulations.

**Figure 6 pharmaceutics-15-02751-f006:**
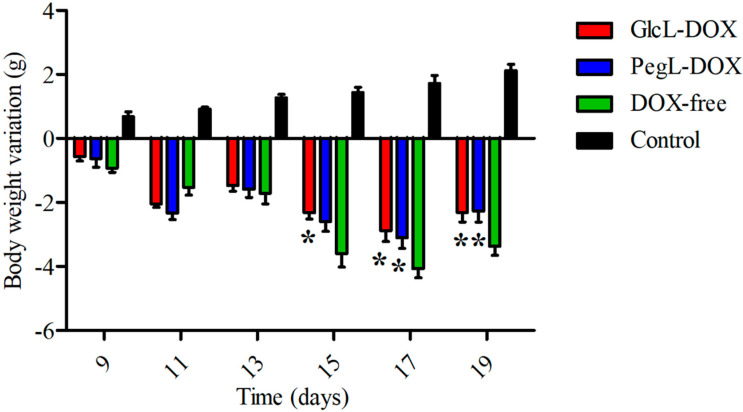
Body weight variation of 4T1 tumor-bearing female mice after treatment with Saline 0.9% (Control group), Free-DOX, PegL-DOX, or GlcL-DOX. Results are presented as the mean ± SD (*n* = 7). All data were analyzed by one-way ANOVA analysis of variance, followed by Tukey’s post-test. * represents statistical differences (*p* < 0.05) between the liposome-treated groups and Free-DOX group.

**Table 1 pharmaceutics-15-02751-t001:** Z-Average diameter, Polydispersity Index (PDI), zeta potential, and Encapsulation Percentage (EP) of Glucose-functionalized liposomes (GlcL-DOX) and Pegylated liposomes (PegL-DOX).

	Z-Average Diameter (nm)	PDI	Zeta (mV)	EP (%)
GlcL-DOX	137.5 ± 5.8	0.10 ± 0.04	−11.0 ± 0.3	100.3 ± 1.7
PegL-DOX	129.8 ± 3.3	0.11 ± 0.01	−2.6 ± 0.4	100.3 ± 4.2

Results are presented as the mean (*n* = 3) ± SD. All data were analyzed by one-way ANOVA analysis of variance, followed by Tukey’s post-test.

**Table 2 pharmaceutics-15-02751-t002:** Half-maximum inhibitory concentration (IC_50_) for Free-DOX, GlcL-DOX, and PegL-DOX against 4T1 tumor cells (*p* < 0.05).

Sample	IC_50_ (µM)
Free-DOX	3.15 ± 0.22
GlcL-DOX	14.50 ± 1.60 *
PegL-DOX	42.35 ± 2.85 *

Results are presented as the mean (*n* = 3) ± SD. All data were analyzed by one-way ANOVA analysis of variance, followed by Tukey’s post-test. * represents statistical differences (*p* < 0.05) between the groups.

**Table 3 pharmaceutics-15-02751-t003:** Relative tumor volume (RTV) and tumor growth inhibition ratio (IR) after treatment with saline 0.9% (Control), Free-DOX, PegL-DOX, and GlcL-DOX.

Group	RTV (Mean ± SD)	IR (%)
GlcL-DOX	3.49 ± 1.05 ^a,b^	58.5
PegL-DOX	5.43 ± 1.46 ^a^	35.3
Free-DOX	6.54 ± 1.29	24.3
Control	8.41 ± 2.56	-

Results are presented as the mean (*n* = 7) ± SD. All data were analyzed by one-way ANOVA analysis of variance followed by Tukey’s post-test. ^a^ represents statistical differences between Free-DOX, GlcL-DOX or PegL-DOX, and the Control group. ^b^ represents statistical differences between GlcL-DOX and PegL-DOX or Free-DOX.

**Table 4 pharmaceutics-15-02751-t004:** Biochemical analyses of 4T1 tumor-bearing female mice after treatment with Saline 0.9% (control group), DOX-free, PegL-DOX, or GlcL-DOX.

	Control	DOX-free	GlcL-DOX	PegL-DOX
ALT (U/L)	21.3 ± 1.8	43.6 ± 14.5 ^a^	32.3 ± 2.6 ^b^	40.1 ± 7.8 ^a^
AST (U/L)	109.4 ± 15.2	121.6 ± 17.5	120.9 ± 10.7	123.8 ± 9.1
Creatinine (mg/dL)	0.46 ± 0.06	0.36 ± 0.03 ^a^	0.35 ± 0.02 ^a^	0.35 ± 0.03 ^a^
Urea (mg/dL)	54.7 ± 10.5	78.8 ± 10 ^a^	64.9 ± 5.4 ^b^	64.8 ± 8.3 ^b^
CK-MB (U/L)	34.7 ± 3.1	42.5 ± 4.7 ^a^	25.6 ± 5.2 ^b^	25.6 ± 6.1 ^b^

AST: aspartate aminotransferase. ALT: alanine aminotransferase. CK-MB: creatine kinase MB isoform. Results are presented as the mean ± SD (*n* = 7). All data were analyzed by one-way ANOVA analysis of variance, followed by Tukey’s post-test. ^a^ represents a statistical difference between the treated group and the control group. ^b^ represents a statistical difference when compared with the DOX-free group.

## Data Availability

Data are contained within the article.

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
