# Peer review of "In Vitro and Preclinical Antitumor Evaluation of Doxorubicin Liposomes Coated with a Cholesterol-Based Trimeric β-D-Glucopyranosyltriazole"

_pharmaceutics, 2023, doi:10.3390/pharmaceutics15122751_

Round 1

Reviewer 1 Report

Comments and Suggestions for Authors

The manuscript compares a PEGylated liposomal doxorubicin to a liposomal doxorubicin construct wherein the surface PEG has been replaced with a glycosyl triazole glucose derivative. The topic is highly relevant, as there is significant interest in exploring the immunogenic effects of repeated PEG use and as well as in reducing/eliminating the use of PEG where possible.

The authors provide an adequate physicochemical characterization of the two materials, including size distribution, zeta potential, encapsulation efficiency, and storage stability. I just ask for two minor corrections/clarifications in this regard:

·        Line 241: I believe the authors mean to say “<0.3”.

·        Table 1: What does Mean Diameter refer to? Is this the Z-Average diameter?

Next the authors perform in vitro and in vivo studies in a relevant 4T1 breast cancer model. The authors should explain why they elected to compare to their PegL-DOX formulation as opposed to comparison to traditional Doxil. To make the argument this new GlcL-DOX formulation is potentially a clinically viable alternative they should have made the direct comparison to current standard of care therapy, Doxil. They do show reduced levels of CK-MB suggesting there would be reduced cardiotox. However, as the introduction points out, the new GlcL-DOX formulation is meant to reduce the potential PEG immunogencity issues associated with Doxil. The manuscript compares a PEGylated liposomal doxorubicin to a liposomal doxorubicin construct wherein the surface PEG has been replaced with a glycosyl triazole glucose derivative. The topic is highly relevant, as there is significant interest in exploring the immunogenic effects of repeated PEG use and as well as in reducing/eliminating the use of PEG where possible.

The authors provide an adequate physicochemical characterization of the two materials, including size distribution, zeta potential, encapsulation efficiency, and storage stability. I just ask for two minor corrections/clarifications in this regard:

·        Line 241: I believe the authors mean to say “<0.3”.

·        Table 1: What does Mean Diameter refer to? Is this the Z-Average diameter?

Next the authors perform in vitro and in vivo studies in a relevant 4T1 breast cancer model. The authors should explain why they elected to compare to their PegL-DOX formulation as opposed to the traditional Doxil formulation. To make the argument this new GlcL-DOX formulation is potentially a clinically viable alternative they should have made the direct comparison to current standard of care therapy. They do show reduced levels of CK-MB suggesting there would be reduced cardiotox. However, as the introduction points out, the new GlcL-DOX formulation is meant to reduce the potential PEG immunogencity issues associated with Doxil; thus, free doxorubicin is not the true clinical comparator for this formulation. If comparative data with Doxil are available, I believe it would enhance the manuscript. If the data are not available, the authors should explain the rationale for choice of their comparator.

Did the authors look at in vitro complement activation of the two formulations?

Author Response

Response letter attached

Reviewer 2 Report

Comments and Suggestions for Authors

The paper presents the preparation and characterization of doxorubicin liposomes coated with a glycosyl triazole glucose. The cytotoxicity and preclinical anticancer effects were also investigated.

This manuscript is generally well structured, and I think it enriches the current research. The work has its advantages, but from my point of view it requires some minor corrections.

Here are some suggestions:

The cytotoxicity of Free-DOX, GlcL-DOX, and PegL-DOX to normal non-cancerous cells should be assessed to exclude their general toxicity.

Text formatting should be carefully checked.

Comments on the Quality of English Language

Minor editing of English language required

Author Response

Response letter attached

Reviewer 3 Report

Comments and Suggestions for Authors

The manuscript "In vitro and preclinical antitumor evaluation of doxorubicin 2 liposomes coated with a cholesterol-based trimeric β-d-GLU-COpyranoSYLtriazole " by E Silva ATM et al. describes a new DOX-loaded carbohydrate-coated liposome prepared and characterized with suitable size, zeta potential, and  storage stability for preliminary biological analyses. It is very interesting and well structured. Interesting subject matter, presentation of the topic legible and the results of the manuscript are very interesting  there's just one little question before the publication on Pharmaceutics.

·         Why was the MTT test only performed at 48 hours?

Best regards

Author Response

Response letter attached
